# Feasibility and Acceptability of a Remote Stepped Care Mental Health Programme for Adolescents during the COVID-19 Pandemic in India

**DOI:** 10.3390/ijerph20031722

**Published:** 2023-01-17

**Authors:** Kanika Malik, Tejaswi Shetty, Sonal Mathur, James E. Jose, Rhea Mathews, Manogya Sahay, Preeti Chauhan, Pooja Nair, Vikram Patel, Daniel Michelson

**Affiliations:** 1Jindal School of Psychology and Counselling, O.P. Jindal Global University, Sonipat 13100, India; 2PRIDE Project, Sangath, New Delhi 110030, India; 3Department of Global Health and Social Medicine, Harvard Medical School, Boston, MA 02115, USA; 4Department of Global Health and Population, Harvard TH Chan School of Public Health, Boston, MA 02115, USA; 5School of Psychology, University of Sussex, Brighton BN1 9RH, UK; 6Department of Child and Adolescent Psychiatry, Institute of Psychiatry, Psychology and Neuroscience, King’s College London, London SE5 8AF, UK

**Keywords:** adolescents, mental health, remote intervention, stepped care, COVID-19, mixed methods, India

## Abstract

Remote mental health services were rapidly deployed during the COVID-19 pandemic, yet there is relatively little contemporaneous evidence on their feasibility and acceptability. This study assessed the feasibility and acceptability of a stepped care mental health programme delivered remotely by lay counsellors to adolescents in New Delhi, India, during a period of ‘lockdown’. The programme consisted of a brief problem-solving intervention (“Step 1”) followed by a tailored behavioural module (“Step 2”) for non-responders. We enrolled 34 participants (M age = 16.4 years) with a self-identified need for psychological support. Feasibility and acceptability were assessed through quantitative process indicators and qualitative interviews (*n* = 17 adolescents; *n* = 5 counsellors). Thirty-one (91%) adolescents started Step 1 and 16 (52%) completed the planned Step 1 protocol. Twelve (75%) of the Step 1 completers were non-responsive. Eight (67%) non-responsive cases started Step 2, all of whom met response criteria when reassessed at 12 weeks post-enrolment. Adolescents favoured voice-only sessions over video-calls due to privacy concerns and difficulties accessing suitable devices. Counsellors noted challenges of completing remote sessions within the allotted time while recognising the importance of supervision for developing competence in new ways of working. Both adolescents and counsellors discussed the importance of working collaboratively and flexibly to fit around individual preferences and circumstances. Disentangling pandemic-specific barriers from more routine challenges to remote delivery should be a focus of future research.

## 1. Introduction

The COVID-19 pandemic has led to severe disruptions and adverse physical and mental health outcomes for communities across the globe, with young people disproportionately affected by interrupted education, social isolation, and limited employment opportunities [1,2]. These impacts have been especially pronounced in India, where COVID- related school closures persisted for almost two years in many states. Despite a shift towards online learning, a large proportion of students from disadvantaged backgrounds struggled to access this provision routinely [3,4]. Similar accessibility barriers have been reported for remote mental health services in India and other low-resource settings [5,6,7,8]. More contextualised evidence is required to address such barriers and shape ongoing initiatives to expand remote mental health care in India and beyond [7,9,10,11,12]. To address this need, the current study piloted a pragmatic approach to reaching adolescents in need of mental health support during the pandemic. This involved adapting an existing school-based stepped care mental health programme, developed as part of the “Premium for Adolescents” (PRIDE) research programme [13,14], so that it could be accessed by adolescents outside of school premises during the pandemic related lockdown. 

The objectives and deliverables of the PRIDE programme (2016–22) were aligned with India’s national adolescent health programme, the Rashtriya Kishor Swasthya Karyakram (RKSK), which was launched in January 2014 with an ambition to reconfigure the prevailing clinic-based health system and instead focus on prevention and early intervention for adolescent health and developmental needs. RKSK specifically highlighted mental health as a public health priority and identified school-based mental health services as a key delivery platform [15]. Previous PRIDE research has shown the effectiveness of an in-person, lay counsellor-led version of a first line transdiagnostic problem-solving intervention (“Step 1”) provided in secondary schools for pupils with elevated mental health symptoms [16]. Non-responders are stepped up to a more tailored intervention (“Step 2”), in which behavioural modules are selected according to the main presenting problem [14,17]. 

Here, we present findings on the acceptability and feasibility of the PRIDE stepped care programme when delivered by lay counsellors in a remote format. The study was conducted among young people from low-income communities where smartphones and other internet-enabled devices, if available at all, are typically shared between family members in households with restricted mobile data and limited broadband access. Given these circumstances, the chosen approach emphasised flexibility such that students could choose between voice or video calls, with supplementary materials distributed electronically and by post. The intervention protocol also incorporated collaborative decision-making to determine the need for stepping up. The findings from this study were intended to shape further innovations in remotely delivered interventions, during the pandemic period and beyond.

## 2. Materials and Methods 

### 2.1. Design

We used a mixed method design that assessed quantitative process indicators of feasibility and deployed qualitative semi-structured interviews to explore adolescents’ and counsellors’ experiences of intervention delivery. We also collected data on indicative outcomes using two adolescent-reported measures. The approvals for this study were obtained from local schools, the Institutional Review Boards of Sangath (the implementing organisation), and Harvard Medical School (the sponsor).

### 2.2. Participants

The sampling frame consisted of 3780 students enrolled in grades 9–12 of four secondary schools serving low-income communities in New Delhi, India. Two of these schools were government-aided public schools, and the other two were charity-aided private schools. There were more girls than boys in the sampling frame, reflecting the fact that two of four partner schools were all-girls schools. None of the schools had pre-existing counselling services, either face-to-face or remote. All four schools were closed to in-person teaching and offered an online curriculum for the duration of the study.

Students within the sampling frame were eligible for the study if they had a self-identified need for psychological support (i.e., expressing a felt need, without being formally assessed for clinical severity), and had access to a voice-only telephone or internet-enabled device (smartphone, computer, or tablet). Participants were also required to be proficient in written and spoken Hindi or English in order to comprehend the intervention materials. Additionally, counsellors were invited to participate in a separate focus group discussion (FGD) at the end of the study.

### 2.3. Measures

#### 2.3.1. Quantitative Feasibility Indicators

We conceptualised feasibility as the extent to which an intervention can be successfully used or conducted in a given setting [18]. Referral logs and electronic case records were used to obtain data on feasibility indicators related to demand (number of referred students); uptake (number of participants who were eligible for and who started each intervention step); completion (number of participants who received the recommended frequency of sessions for each intervention step); reasons for discontinuation; and dosing (the frequency/length of sessions and overall duration of each step). 

Additionally, the quality of intervention delivery was assessed for 20% of the audio-recorded sessions, selected randomly. These recordings were rated independently by fellow counsellors and a supervising psychologist during group supervision meetings, using a 20-item scale that rated therapeutic skills according to the level of demonstrated competence (1 = limited, 2 = basic, 3 = good, 4 = advanced) (see Appendix A for a copy of the rating scale).

#### 2.3.2. Qualitative Individual Interviews and Focus Group Discussion

Semi-structured individual interviews with intervention participants were used to explore the acceptability of intervention content, materials, and delivery processes. A topic guide was constructed in which acceptability was conceptualised as a multi-faceted construct that “reflects the extent to which people receiving a healthcare intervention consider it to be appropriate, based on anticipated or experienced cognitive and emotional responses to the intervention” [19]. A separate FGD was conducted with counsellors, focusing on barriers and facilitators to implementing the intervention. 

#### 2.3.3. Clinical Outcomes 

We used two adolescent-reported measures that have previously been translated into Hindi and used in the local context [20,21]. The Revised Children’s Anxiety and Depression Scale (RCADS-25) is a widely used measure of internalising problems that includes both anxiety and depression subscales [22,23]. The Youth Top Problems (YTP) [24] is an idiographic measure that asks respondents to identify, rank, and rate the severity of three prioritised psychosocial problems. The RCADS-25 and YTP were administered remotely by research assistants at baseline (T0) and 12 weeks later (T2). Additional YTP ratings were obtained by counsellors for progress monitoring and to identify non-responders at the end of Step 1 who may require another Step 2 intervention (see Procedures for further details). Non-response was defined as achieving less than 50% improvement in the severity of the first-ranked priority problem (compared with baseline), and/or no improvement in any of the other two problem ratings.

### 2.4. Procedures

Study referrals were generated between November 2020 and March 2021 using a variety of recruitment materials (see Table 1). 

Research assistants followed up with self-referred students through a voice call and assessed them individually for eligibility using a short screening proforma. The proforma included questions related to the school and grade in which they were enrolled, self-reported difficulties with reading and writing, access to a phone or digital device, and a single item adapted from the Strengths and Difficulties Questionnaire Impact supplement [26]. The latter assessed felt need for psychological support (“Overall, do you think that you have difficulties in any of the following areas for which you need support: emotions, concentration, behaviour or being able to get on with other people?*”*). Verbal assent was recorded for all eligible adolescents (or consent for adolescents aged 18 years and above), followed by recorded verbal parent/guardian consent for those aged less than 18 years. 

Following enrolment, participants were sent printed reference copies of the RCADS-25 and YTP in their preferred language (English or Hindi) by post and electronic copies (non-editable PDF format) through WhatsApp. A research assistant subsequently made telephone contact to assist with administration. After receiving instructions from the research assistant, participants read aloud each item and their corresponding response. Categorical items/scores corresponding to the verbal responses were marked in real-time by the research assistants on the relevant electronic forms using a hand-held tablet device. 

Following the baseline assessment (T0), participants were offered up to five sessions of a problem-solving intervention (Step 1) delivered through a voice or video call. Sessions were intended to last for up to 30 min over a period of 3–4 weeks. The intervention structure was closely based on the in-person version of Step 1 [13], where session 1 introduced a three-part problem-solving technique based around the acronym POD (“Problem”, “Option”, “Do it”); sessions 2–4 supported the applied use of this approach to a prioritised problem; and an optional session 5 was used to consolidate learning and help with generalising the approach to other problems. 

Progress was monitored regularly through in-session ratings on the YTP. Step 1 was concluded at session 4 or 5 for those individuals who showed 50% or greater improvement on the main target problem, accompanied by a downward trend on at least one of their other YTP problem scores. Participants who did not meet response criteria on the YTP by the fourth session were considered for Step 2 through a shared decision-making protocol that was structured around the SHARE acronym [27]. This entailed “seeking” the participant’s involvement by explaining the meaning and purpose of shared decision-making, followed by “helping” the participant to review their progress and exploring the potential costs/benefits of concluding/continuing with counselling. Next, the participant’s preference and justification were “assessed”, after which the counsellor shared their recommendation. The participant was encouraged to “reach” a decision about whether or not to step up. The participant was asked to “evaluate” their satisfaction (on a three-point scale from very satisfied-somewhat satisfied-not at all satisfied) with the final decision, followed by further discussion as needed. A similar shared decision-making process was used to decide on the need for early discharge during Step 1. If a participant achieved early response by session 2 or 3, they could then collaboratively decide with the counsellor to end the intervention at that point.

Those progressing to Step 2 received up to 6 sessions over voice or video call, which were intended to last 30–35 min each and were scheduled over 4–6 weeks. Based on an existing intervention protocol [17], an introductory session on relaxation skills was provided to all Step 2 participants, followed by one of three behavioural modules (2–4 sessions) selected according to the main presenting problem, i.e., behavioural activation for depression, exposure for anxiety, or assertiveness and communication training for conduct problems. The final session focused on relapse prevention. 

In both steps, participants additionally received supplementary resource materials in both printed and electronic (editable PDF) formats. These materials included comic books, which used illustrated stories to describe common problems in the target age group, explain the essential concepts of the various behavioural skills, and offer practical suggestions (“quick tips”) for using these skills effectively. The stories were followed by suggested home practice exercises to develop skills further. Participants were also presented with summary posters at the end of each step.

Both intervention steps were delivered by the same five lay counsellors, who were employed by Sangath. The counsellors had at least two years of experience in delivering Step 1 in a face-to-face format. They received a one-day “top up” office-based training from psychologists (SM, PN, and RM) with a particular emphasis on competencies needed to deliver Step 1 remotely, and a 6-day training period, focused specifically on shared decision-making, and Step 2 (which the counsellors had not previously delivered in any format). Additionally, the counsellors took part in weekly group supervision for the study duration, where audio-recorded sessions were reviewed and rated for quality by peers and a supervising psychologist. The format of group supervision is elaborated elsewhere [20]. For the current study, the training and supervision were conducted in a hybrid mode (partially online and partially in-person) in line with COVID-19 safety protocols.

The YTP was administered by research assistants at enrolment (T0), during Step 1 by counsellors (T1), and again by researchers approximately 12 weeks from study enrolment (or immediately after Step 2 if this extended beyond 12 weeks) (T2). The RCADS was administered by researchers at T0 and T2 only. After the completion of outcome assessments, participants were invited to take part in semi-structured individual interviews. Those who agreed (*n* = 17) were interviewed within two weeks of completing the final outcome assessment. Additionally, a separate FGD (*n* = 5) was held with counsellors once the intervention delivery phase had ended. The assessments, interviews, and FGD were conducted by research staff who were not involved in intervention development or delivery.

### 2.5. Analysis

Quantitative feasibility indicators were examined descriptively using frequencies, means and SDs. Descriptive analysis of clinical outcomes involved comparisons of mean YTP scores (based on the sum of the three problem severity scores) for participants who completed assessments across all three time points (T0, T1, and T2). The YTP score from the last attended Step 1 session was used for T1. On the RCADS, we noted descriptive trends over time in mean T-scores. The T values are standardised scores, with a mean of 50, and the standard deviation of 10. These were calculated from raw scores using spreadsheets available from the developer (link: https://www.childfirst.ucla.edu/resources/ (accessed on 15 November 2020)).

Qualitative data were analysed using a thematic framework approach [28]. Two researchers (KM and TS) independently read all the transcripts in the original Hindi language. A preliminary hierarchical coding framework was then prepared deductively by drawing on the research questions and established conceptual definitions of feasibility and acceptability [18,19]. Initial deductive coding focused on thematic categories of intervention coherence (the extent to which participants understand the rationale for the intervention and what is required of them), affective attitude (how participants feel about an intervention and its elements), effectiveness (perceived changes in valued outcomes that may result from participation), self-efficacy (confidence that participants can do what is required of them), and burden (the perceived amount of effort to take part), and barriers to deliver remotely. 

The preliminary framework was refined in discussion with the senior author (DM), then applied to a subset of transcripts. This was refined iteratively through data-driven coding, with granular codes grouped under higher-order themes and sub-themes. Coded data were organised into a spreadsheet matrix, with participants organised into rows, and theme and sub-themes organised as columns. Quotes from individual interview and FGD transcripts were placed in corresponding cells. Iterative revisions in the framework were reviewed with the senior author. The final stage involved operationalising themes and sub-themes and comparing them across participants to develop a narrative interpretation. Illustrative quotes were selected to supplement narratives and were translated from Hindi to English for the purpose of reporting. 

## 3. Results

### 3.1. Feasibility Indicators

A total of 755 students from 52 classes were sensitised remotely during the study period, and 56 (7%) were referred into the programme. As shown in Figure 1, among those referred, 47 (84%) were assessed for eligibility and 41 (87%) were eligible. Out of 41 eligible adolescents, 34 (83%) completed the assent/consent and baseline assessment procedures (T0). The demographic and clinical characteristics of these 34 participants were as follows: M age = 16.4 years, SD = 1.6; females *n* = 25 (74%); M RCADS Total T-score = 62.4, SD = 14.2; and M YTP score = 7.5, SD = 1.8. 

All 34 participants who enrolled in the study opted to have sessions through voice-only calls. From this group, 31 (91%) started Step 1, and 15 (48%) completed the intended number of Step 1 sessions (4 or 5). Additionally, one participant was offered early discharge (at session 3) and counted as completer, based on criteria mentioned above. Among those participants completing Step 1, 12 (75%) participants were non-responsive. Eight (67%) out of these 12 non-responders started Step 2, three non-responders opted out, and one was removed from the study and referred to a mental health professional for management of high suicidal risk. Demographic characteristics of Step 2 participants were as follows: M age = 16.0 years, SD = 1.4; females *n* = 6 (75%). All but one of the eight Step 2 participants completed the full course of Step 2. The reasons for discontinuing each intervention step are outlined in Figure 1.

Step 1 completers attended, on average, 4.6 sessions (SD = 0.8), spread over 31.1 days (SD = 24.1). Step 2 completers attended an additional 5.1 sessions (SD = 0.7), spread over 54.4 days (SD = 18.5). The mean duration of Step 1 and Step 2 sessions was 45 min (SD= 11.0) and 59 min (SD = 11.20), respectively. Those who completed both intervention steps attended a mean of 9.1 sessions (SD = 0.7), spanning 84 days (SD = 23.0). Quality assessments of intervention delivery were in the “good” to “advanced” range for both steps and across both peer and supervisor scores (mean supervisor rating, Step 1 = 3.6, SD = 0.5; mean peer rating, Step 1= 3.8, SD = 0.2; mean supervisor rating, Step 2 = 3.4, SD = 0.3; mean peer rating, Step 2 = 3.8, SD = 0.2).

### 3.2. Qualitative Findings

Seventeen adolescents took part in individual interviews (M age = 16.3 years, SD = 1.3; females *n* = 12 [71%]; received Step 1 only, *n* = 10 [59%]; received both Step 1 and 2, *n* = 7 [41%]). Five counsellors (M age = 31.2 years, SD = 4.1; females, *n* = 3 [60%]) took part in a separate FGD. Five overarching themes were developed: (i) coherence of the intervention steps; (ii) usefulness of counsellors’ guidance and supporting materials; (iii) balancing structure and adolescents’ needs in shared decision-making; (iv) valued outcomes and skill development; and (v) implementation of remote delivery methods. These are elaborated below, illustrated with relevant quotes.

#### 3.2.1. Coherence of the Intervention Steps

Problem solving was viewed by all adolescents as a useful skill. Interviewees, including some participants who dropped out early from Step 1, frequently recalled the “POD” acronym and described its importance in facilitating the process of problem solving, 


*“Ma’am, I really liked ‘POD’. Through this, I learned recognising problems, then finding their options and learning to do it.”*
(Adolescent P9, 15 years, female, dropped out from Step 1)

Adolescents who took part in Step 2 found it beneficial to learn additional complementary behavioural skills to help with their persisting problems. While all acknowledged the benefit of Step 2 skills, there was mixed feedback about the appropriate placement of problem-solving skills in the intervention framework. Some interviewees acknowledged the benefit of learning broad problem-solving skills before the problem-specific Step 2 behavioural skills. 

*“I prefer learning problem solving first. To me, being active* [the Step 2 behavioural activation module] *looked like one option of broad problem-solving framework. So, we should learn problem solving first and then gradually we can learn different options as part of this skill.”*(Adolescent P6, 15 years, female, completed both steps)

Others felt that personalised Step 2 skills (i.e., matched to specific problem types) should be taught earlier to accelerate therapeutic gains.

*“Deep breathing and overcoming fear* [the Step 2 exposure module] *have become my favourite* [compared to POD]. *Till the fourth or fifth session,* [counsellor] *sir taught us about POD and these other skills were taken up in the sessions after that. It would have been even better for our progress to learn things like deep breathing and overcoming fear much earlier.”*(Adolescent P3, 16 years, male, completed both steps)

#### 3.2.2. Usefulness of Counsellors’ Guidance and Supporting Handouts

Counsellors were praised by adolescent interviewees as supportive guides who offered helpful suggestions for solving problems when participants felt stuck. 

*“Madam* [the counsellor] *was very understanding and quickly understood everything I shared with her. She helped me a lot in finding alternative solutions to problems. Initially, I felt very nervous, but madam told us how to proceed and gave us step-by-step directions and tips on finding solutions, which helped me greatly.”*(Adolescent P7, 16 years, female, completed Step 1 and opted out of Step 2)

Adolescents also appreciated counsellors’ willingness to adapt the pace of sessions and work collaboratively, especially when deciding about whether to continue beyond Step 1.

*“She* [counsellor] *neither give me any suggestions, nor did she pressurise me to continue or terminate. She had left it for me to decide, i.e., if I wanted to end, I could end it, and if I wanted to continue, then I could continue also; that was my choice. This was a good thing.”*(Adolescent P6, 15 years, female, completed both steps)

In line with adolescents’ feedback, counsellors also recognised the importance of pacing sessions according to individuals’ needs and preferences.


*“If we sense a student is reluctant to open up or talk about his difficulties, we try to take it slow. We talk about their interest areas, things they like and ask more specific questions to help them express themselves more concretely. We will provide them with step-by-step guidance if they struggle to open up or express their concerns. Usually, by the end of a couple of sessions, they feel more comfortable with the process. By the time they are in Step 2, they are pretty vocal and engaged in the process.”*
(Counsellor C2, female)

Adolescents reported that the comic books and posters helpfully complemented counsellors’ inputs by keeping them engaged in between sessions, while also serving as a source of ready-made solutions for solving problems. Suggestions were also made about expanding the repertoire of stories to cover a wider array of problems and potential solutions. 

*“The most helpful thing in both these booklets was that after reading their story, it seemed that when they* [story characters] *can* [face their fears], *why can’t I. The boy in the story feels scared of dogs and no one else feels that way; similarly, I feel scared of certain things, but others don’t. This is something we can learn and correct. Then, I did the right thing after reading the booklet.”*(Adolescent P3, 17 years old, male, completed both steps)

*“The stories given were very good, as you can understand everything from the story. Then some ready-made solutions have been given, which are also very good. You people* [programme developers] *can write more stories... like something based on my problem. My main problem is overthinking; you can make a story about it and then you should give me that booklet.”*(Adolescent P14, 15 years old, female, completed both steps)

#### 3.2.3. Balancing Structure and Adolescents’ Needs in Shared Decision-Making

In line with adolescents’ feedback, counsellors noted the importance of flexible pacing of sessions to accommodate individuals’ learning pace and time constraints to competing demands. At the same time, counsellors spoke positively about the structured intervention manual which detailed the sequence of in-session activities. Supervisory feedback was especially useful for implementing novel tasks from the updated manual. 

*“Initially, it was challenging to do the shared decision-making. We struggled with how to say it, frame it in age-appropriate language, explain why* [the adolescent] *needs to get involved in it, when we should raise this topic, and what the next steps will be. To overcome these challenges, we used to read* [the step-by-step manual], *and discuss this thing with other counsellors. Then we also did role-play. Our supervisor also helped. It was a bit of a challenge initially, but all this made it easier as we moved ahead.”*(Counsellor C5, female)

While most adolescents were satisfied with how the shared decision-making process was structured by the counsellors, a few suggested that more information was needed. 

*“Yes, it would have been good if she* [counsellor] *had explained what will happen next in the counselling journey. For example, had I known the total number of sessions* [in Step 2], *I would have taken the decision instantly rather than taking too much time to think about it. But I was confused about how long the counselling will go on. Also, she told me the benefits of continuing counselling but not the drawbacks of ending counselling. When you* [counsellor] *are helping me, explain both benefits and drawbacks so that I can be more confident in making the decision. Ultimately, it will be my decision only, but it will be good if I get more information and help from the counsellor.”*(Adolescent P2, 16 years old, female, completed both steps)

#### 3.2.4. Valued Outcomes and Skill Development

Adolescent interviewees described improvements across a wide range of symptoms of anxiety, depression, and anger-related problems, and functional domains, such as academic performance, interpersonal difficulties with family and peers, and difficulties related to time management. Many of the adolescent participants described how their problems had benefited from the application of problem-focused coping skills from Step 1 and additional behavioural skills learned from Step 2. 


*“During the sessions, the counsellor made me complete a graph that helped me understand how my problems were going. Then counsellor asked me to work on it. Based on our discussion, I worked on generating and applying alternative solutions, which helped reduce problems.”*
(Adolescent P4, 18 years old, male, completed Step 1 and opted out of Step 2)

In addition to relief in problems for which they sought counselling, participants reported changes in their overall approach to managing problems, including realising the importance of making an active effort to solve problem and the need for continued practice. 


*“Earlier I used to procrastinate but now I’m much faster. The counselling helped me learn that we have to plan and take steps to solve problems, troubles won’t go on their own.”*
(Adolescent P14, 18 years old, male, completed Step 1 and opted out of Step 2)

*“I made good improvements in counselling like my problem with overthinking were reduced, my fear subsided, and those negative bad thoughts were also gone down. At times, I still get bad thoughts, but now I can manage them. The most important thing is that I have learned to control, no matter what the situation is... Earlier, if I was afraid to go somewhere, I would not go, but now I try to control, deep breathe, plan to face* [my fear] *and go wherever I’m needed.”*(Adolescent P3, 17 years old, male, completed both steps)

Some participants recognised the limits of what could be achieved through the problem-solving approach, particularly with interpersonal issues in the context of pandemic.

*“We talked about* [my problem with] *trust issues, and came out with a few options, but I did not feel there was any suitable option, given the lockdown constraints. I concluded that leaving it as it is okay, and we do not need anything… Honestly, not much could be done about this but to leave it aside.”*(Adolescent P9, 17 years old, female, completed both steps)

#### 3.2.5. Implementation of Remote Delivery Methods

Adolescents and counsellors held mixed opinions about attending counselling remotely. Some adolescents felt more at ease with the format of telephone sessions, as opposed to the prospect of in-person or video meetings. 

“[Counselling] *was very good on the phone because sometimes it happens with me that I cannot tell everything in detail face-to-face, but on the phone, it was easy to tell… If you are face-to-face, then I keep thinking about my and the counsellor’s facial expressions, which makes me feel a little hesitant.”*(Adolescent P16, 18 years old, female, completed/responded to Step 1)

Counsellors reported that remote counselling afforded students a larger time window to schedule sessions, thus providing options for timetabling outside of conventional school hours. Counsellors also appreciated being able to use WhatsApp for sending reminders to students about forthcoming sessions, and for sharing electronic copies of intervention materials where needed. Notwithstanding these benefits of remote communication, there were also frequent reports of accessibility issues, such as students having to share telephones with family members, lack of privacy at home when taking part in sessions, and poor connectivity. These issues ruled out video calls for all participants and impacted the overall continuity and duration of sessions. Counsellors often needed to extend sessions to compensate for the extra time incurred in explaining and practising skills over the telephone amidst various disruptions. 


*“Sometimes our sessions extend beyond an hour, but that is primarily due to network and connectivity issues. With multiple call drops, the counsellor had to wait 10–15 min before we could get into stable connectivity.”*
(Adolescent P2, 16 years old, female, completed both steps)

*“As everything was new* [for students] *in remote counselling, we had to set the ground rules and agenda, which took longer than we planned… A lot of our time was spent monitoring the progress, especially in the first session; we had to tell students how to do the progress monitoring for each problem. This was very difficult to explain on the phone, the child would be confused and since we could not see them, we just relied on whatever they said…Even other activities in booklets used to take more time, as we were remote, we had to explain to them go on this page, do this here and then go on that page,* etc. *This used to take half of the session, and then we will reach the main session agenda and to complete them well, it was necessary to spend time there too. So, this is why the session used to go on for longer than what was given in the manual.”*(Counsellor C3, Male)

Consequent to these accessibility issues, most young people favoured a face-to-face format for future rounds of counselling.

“[Counselling] *was good on the phone, as my problem is almost resolved. However, it would have been much better if she* [counsellor] *was offline. This entire year, whether it was counselling or the school classes, it was online, and we do not feel so comfortable with it. Talking face-to-face would have been much better than online. The counsellor will also find it much easier to work out solutions for our problems. Even from our body language, she can observe the smallest of things, which would help her understand us much better.* [Even on the phone] *she did understand everything, but it would have been better if it was offline.”*(Adolescent P7, 16 years old, Female, completed/responded to Step 1)

Notwithstanding feasibility concerns, counsellors remarked on the high motivation of many participants and the potential to overcome initial trepidation and establish positive therapeutic relationships at a distance. 

*“We got one advantage* [due to COVID restrictions] *that 2–3 children out of every five children who came to us were either feeling bored at home or were in a situation where they could not talk to others and express themselves. So, when they came to us, we spent time building trust and genuinely showed concern for them during our session; they liked it a lot. They got engaged, thinking this was a platform where they could talk and express their viewpoints. They intrinsically felt motivated about the sessions and then we didn’t have to put in too much effort… Once the rapport is well-established, they easily open up and start talking about their problems.”*(Counsellor C3, male)

### 3.3. Indicative Outcomes

Complete outcome data on the YTP and RCADS were available for 27 (79%) out of 34 participants who enrolled in the study. As shown in Figure 2, “Category 1” participants who responded by the end of Step 1 experienced further improvements in problem scores up to 12 weeks, with all meeting the response criteria at T2. Among the participants who were non-responsive at T1, those who opted out of receiving any further intervention (“Category 2”), and those who continued to Step 2 (“Category 3”) followed similar trajectories of improvement in YTP scores. Except for one participant in each of Categories 2 and 3, all met response criteria at T2. This contrasted with the smaller reduction in YTP scores from T1 to T2 seen among “Category 4” participants who dropped out before completing Step 1. The majority (7 out of 12) of these participants did not meet response criteria at 12-week follow-up.

The mean baseline RCADS T-score for the 27 participants with paired pre-post scores was 62.4 (SD = 14.2). Follow-up RCADS scores were similar across each of the four categories: Category 1 (M = 44.8, SD = 9.5), Category 2 (M = 44.4, SD = 4.1), Category 3 (M = 44.6, SD = 9.8), and Category 4 (M = 44.8, SD = 15.8).

## 4. Discussion

This study examined the feasibility and acceptability of a remotely delivered stepped care programme for common mental health problems among young people in India during the COVID-19 pandemic. We observed a limited uptake of the remote programme in the targeted population of secondary school pupils during the period of COVID-19 related school closures, with enrolment mostly restricted to older adolescents (mean age = 16.4 years). The non-completion rate for the first-line intervention was also higher than in earlier studies, conducted before the pandemic, in which we delivered a similar intervention on school premises [16,17]. Given a choice of different access options for the remote counselling, participants uniformly chose voice-only calls over video-calls, due to privacy concerns, and a lack of internet-enabled devices. Poor connectivity during calls also played a part in longer sessions than planned.

Our findings align with other evidence on demographic and accessibility related barriers to remote mental health service delivery in low-resource settings, before and during the pandemic [29,30,31,32,33,34]. We sought to overcome anticipated feasibility issues by keeping technological requirements of the stepped care programme to a minimum (e.g., participants could access counsellors using a basic mobile phone), by offering free airtime, and by scheduling sessions flexibly to fit around online schooling. Counsellors’ transition to working remotely was facilitated by a structured intervention manual and regular supervision, and intervention quality ratings were generally at the higher end of the scale. Although a number of engagement challenges remained, qualitative interviews revealed that adolescents valued the available support from counsellors, and the behavioural focus was highly relevant to coping with stressors during the pandemic. 

We note the relatively low response rate (25%) for participants immediately after completing Step 1, compared with response rates of around 50% in previous evaluations of Step 1 when delivered in-person [13,16,17]. Interpretation of outcomes is further complicated by differences in response criteria between studies, and the relatively high proportion of participants (48%) in the current study who had an unplanned ending to Step 1. Unlike the previous round of studies, we did not observe rapid or sustained improvement among participants who stopped attending Step 1 [13,16]. The lack of complete data on reasons for dropout, and potential demand characteristics affecting the data that was available, prevent any firm conclusions about why participants dropped out. However, it seems plausible that in many cases this was due to accessibility or acceptability reasons rather than a reduced need for psychological help.

Outcome scores were available for eight participants who entered Step 2, and majority of them met the response criteria at the final follow-up assessment. The very small number of Step 1 non-responders who opted out from Step 2 prevents meaningful comparison of outcome trajectories, but our qualitative findings offer corroborative evidence for the incremental benefits of Step 2. Positive experiences of shared decision-making were also reflected in the qualitative data. This collaborative decision-making process may have contributed to a higher opt-in rate for Step 2 compared with an earlier evaluation of stepped care in which stepping up decisions were based solely on outcome scores [17]. More robust research designs will be needed to confirm this inference. 

This research sits alongside another study by our group [8], which highlighted the feasibility challenges of implementing a randomised controlled trial of an online problem-solving intervention for young people during the COVID-19 pandemic in India. The external validity of our findings is further strengthened by adopting a pragmatic approach to remote delivery and the triangulation of multiple data sources and perspectives, although we recognise that the small sample size and lack of a control group prevents conclusions about intervention effectiveness. We also recognise that many of the feasibility barriers described in this study will have been exacerbated by the pandemic, which limits wider applicability of findings. For example, access to telephones/computers would have been curtailed by demand from other household members in lockdown, while privacy would have been even more difficult to achieve in crowded homes. 

We note that the current study was initiated in the first year of the pandemic, prior to the inauguration of the government-funded National Tele-Mental Health Program (NTMHP) in 2022. Future research is needed to disentangle pandemic-specific barriers from more routine accessibility barriers affecting use of remote mental health services in large-scale, nationwide initiatives arising from NTMHP in India [11], and comparable developments in other low-resource settings [12]. Furthermore, studies should investigate infrastructure and resources that may reduce long-standing barriers. Examples could include subsidies and waivers for purchasing digital devices and airtime packages for vulnerable populations. Offering a wider choice of remote intervention formats (e.g., online, tele-counselling, and hybrid/blended approaches) may also help with feasibility and acceptability. In this direction, research should explore preferences and priorities of end users and other relevant stakeholders (parents, service providers, school authorities, and technology/communication companies) for remote options across a variety of therapeutic modalities, including those such as family therapy which conventionally use conjoint or group formats. Such evidence is important for ensuring that practice innovations arising from the pandemic do not perpetuate inequalities in access for historically disadvantaged groups.

## 5. Conclusions

The current evaluation adds to the limited research literature on remote mental health interventions for young people in low- and middle-income countries. A stronger emphasis on evidence-informed, contextually appropriate innovations is needed to narrow existing gaps in accessibility and widen the proportion of young people who stand to gain from the use of communication technologies for mental health service delivery. In the meantime, the growing enthusiasm for a “digital first” approach to service delivery accelerated during the COVID-19 pandemic must be tempered by on-the-ground realities.

## Figures and Tables

**Figure 1 ijerph-20-01722-f001:**
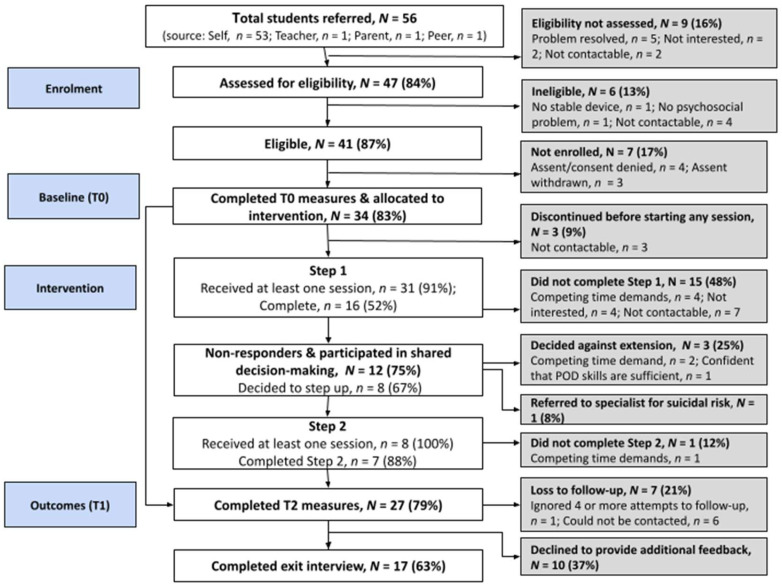
Participants flow in research and intervention activities.

**Figure 2 ijerph-20-01722-f002:**
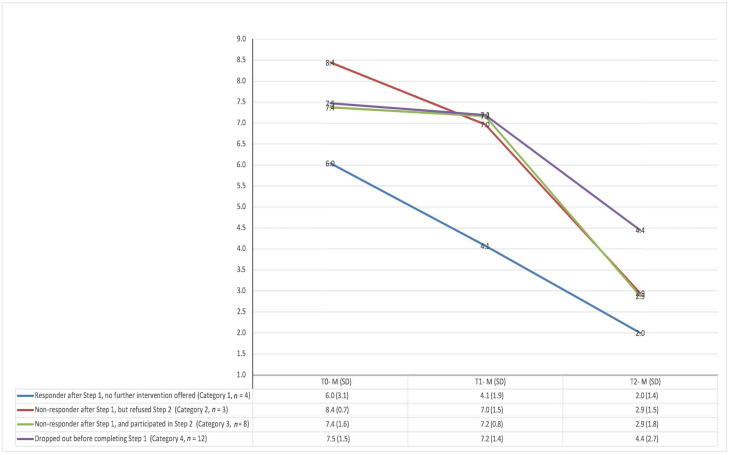
Youth Top Problem (YTP) average scores across three time points.

**Table 1 ijerph-20-01722-t001:** Recruitment activities and materials.

Activity/Material	Description
Student webinar	Adapted from an existing in-person classroom ‘sensitisation’ session [25]. Attendees were presented with an animated video and a PowerPoint presentation that contained age-appropriate information about common mental health problems and explained the rationale for the available intervention. This was followed by a counsellor-facilitated group discussion about the practicalities of remote delivery and procedures for enrolling in the study. Separate webinars were offered to each class.
Teacher webinar	Structured around a similar PowerPoint presentation as the student webinar, followed by a group discussion concerned with how teachers could effectively disseminate information about the study and encourage students to participation in the intervention.
Digital flyer	Contained a brief description of the available intervention and how to enrol in the study. This was circulated by teachers to class WhatsApp groups. Students were able to begin the enrolment process either by calling or messaging (WhatsApp) a designated number without incurring any airtime charges.
YouTube video	Contained an abbreviated version of the same information that was presented by counsellors during the webinar, in which a member of the study spoke to camera. It was intended for students who could not attend the live webinar due to technical issues or competing activities. Teachers circulated this to class WhatsApp groups along with the flyer.

## Data Availability

The raw data and coding framework used in this study can be accessed on request from the corresponding author.

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
