# Peer review of "Feasibility and Acceptability of a Remote Stepped Care Mental Health Programme for Adolescents during the COVID-19 Pandemic in India"

_ijerph, 2023, doi:10.3390/ijerph20031722_

Round 1
Reviewer 1 Report
As a feasibility and acceptability study, this is a small pilot. The study design is elegant and robust. In a country where the MhGap issue has been highlighted often, even before the pandemic, such studies would help in scaling up processes.
Since the submission is to a journal addressing public health, the discussion needs to focus on public health implications in more detail. In its current form, the paper has a disproportionate focus on the qualitative data. While acknowledging that this focus is relevant, its application in existing systems of care is hardly alluded to.
There are two examples of this. The Rashtriya Kishore Swasthya Karyakram (RKSK), a flagship program of the National Health Mission in India is operational. How does this pilot contribute to the vitalizing of this public health initiative for adolescents?
Acknowledging the mental health crisis in wake of the COVID-19 pandemic and an urgent need to establish a digital mental health network that will withstand the challenges amplified by the pandemic, Government of India announced National Tele Mental Health Programme (NTMHP) in the Union Budget 2022-23.
Specialised care is being envisioned through the programme by linking Tele-MANAS with other services like National tele-consultation service, e-Sanjeevani, Ayushman Bharat Digital Mission, mental health professionals, Ayushman Bharat health and wellness centres and emergency psychiatric facilities. Eventually, this will include the entire spectrum of mental wellness and illness, and integrate all systems that provide mental health care.
Since the paper is about remote care, the discussion needs to examine the results in the light of their significance in a massively funded remote care program.
Reviewer 2 Report
Very interesting experience with vulnerable adolescents in India. It is very important to investigate the possibilities that technology offers for remote mental health services. However, I propose attention to the following issues:
1. Reflection on communication with parents and families of adolescents. How do you interpret family therapy? is it possible remotely?
2. Reflection on privacy and the therapeutic alliance among the participating adolescents and the counselors. Many questions arise for the reader of the text.
3. How have you evaluated the social determinants in the intervention with adolescents?
4. Psychosocial data? where are they? feeding? Physical Health? social health?
5. And the blended therapies? that combine face-to-face and online communication? do you know this?
6. More information about the directors? They were all psychologists.
7. Explain in detail the results of the focus group on barriers and assessments?
8. Present quantitative data in tables and quantitative data synthesized from Grounded Theory.
9. A deeper reflection on the limitations of the investigation. A more critical appraisal. Evaluate the optimistic and pessimistic positions.
10. Is the small size of the sample sufficient to sustain that the attention provided to adolescents in the participating schools is feasible?
11. The research you have developed is highly commendable, I congratulate you. It is obvious that more research is needed to advance online mental health services for vulnerable social groups. But this is a very general conclusion; they must specify and critically assess the social and cultural obstacles and barriers identified in their research.
12. I recommend looking for academic literature on similar experiences in developing or developed countries, to compare and discuss results.
Round 2
Reviewer 2 Report
Thank you very much for the answers. I encourage you to move forward with the research and the questions raised.